# Surface Functionalization Strategies of Polystyrene for the Development Peptide-Based Toxin Recognition

**DOI:** 10.3390/s22239538

**Published:** 2022-12-06

**Authors:** Ahmed M. Debela, Catherine Gonzalez, Monica Pucci, Shemsia M. Hudie, Ingrid Bazin

**Affiliations:** 1HSM, University Montpellier, MT Mines Ales, CNRS, IRD, Ales, 30119 Ales, France; 2LMGC, University Montpellier, IMT Mines Ales, CNRS, Ales, 30119 Ales, France; 3Department of Chemistry, National Taiwan University, Taipei 10617, Taiwan

**Keywords:** polystyrene, surface modification, coupling agents, ochratoxin A, food monitoring, water monitoring

## Abstract

The development of a robust surface functionalization method is indispensable in controlling the efficiency, sensitivity, and stability of a detection system. Polystyrene (PS) has been used as a support material in various biomedical fields. Here, we report various strategies of polystyrene surface functionalization using siloxane derivative, divinyl sulfone, cyanogen bromide, and carbonyl diimidazole for the immobilization of biological recognition elements (peptide developed to detect ochratoxin A) for a binding assay with ochratoxin A (OTA). Our objective is to develop future detection systems that would use polystyrene cuvettes such as immobilization support of biological recognition elements. The goal of this article is to demonstrate the proof of concept of this immobilization support. The results obtained reveal the successful modification of polystyrene surfaces with the coupling agents. Furthermore, the immobilization of biological recognition elements, for the OTA binding assay with horseradish peroxidase conjugated to ochratoxin A (OTA-HRP) also confirms that the characteristics of the functionalized peptide immobilized on polystyrene retains its ability to bind to its ligand. The presented strategies on the functionalization of polystyrene surfaces will offer alternatives to the possibilities of immobilizing biomolecules with excellent order- forming monolayers, due to their robust surface chemistries and validate a proof of concept for the development of highly efficient, sensitive, and stable future biosensors for food or water pollution monitoring.

## 1. Introduction

Utilization of polymer-based materials is continually evolving due to their cost and ease of mass production. Chemical functions of polymer supports allow the widening of their versatility in terms of compatibility, stability, and polarity [1]. The addition of these features will be appealing for applications in fields including biomedicine, microelectronics [2], and the textile industry [3]. Among the various polymer supports available, polystyrene (PS) has more applications in biomedical fields due to its optical transparency, good mechanical properties, and non-toxicity [4,5]. Controlled chemical and/or physical treatment of PS substrates leads to modified surface properties suitable for introducing chemical moieties. The treatment allows the utilization of groups with designed functionality to graft the surface with biomolecules. Consequently, a variety of methods have been reported for modification of PS substrates, including treatment with either piranha solution [6], irradiations with plasma [7], gamma [8], UV-Ozone [9], or electron beam [10]. In addition, the combination of piranha treatment and UV irradiation [11]; KOH treatment followed by O_2_ plasma [12]; and Ozone and UV irradiation in aqueous ammonia [13] are also common methods to oxidize PS substrates, usually resulting in OH and NH_2_ terminated surfaces for further functionalization. Consequently, a significant amount of literature has emerged to tailor surface properties of PS substrates for further activation with coupling agents. Hydroxyl terminated PS substrates are often functionalized with silicates/siloxanes [11,12]. There are limited reports on the use of activated PS to functionalize with cynogene bromide, carbonyl diimidazole, and divinyl sulfone [14,15]. The afore-mentioned coupling agent has a wider use in the functionalization of OH groups in sugar [16]; OH functionalized titanium oxide nanotube [17]; quartz/silica surfaces [18]. These coupling agents are known to offer highly reactive surfaces with the potential to anchor proteins, short peptides, and DNAs. The immobilization method chosen most of the time correlates with biomolecule activity, reproducibility, stability, and limits of detection of the biodetection strategy [19,20].

Ochratoxins are mycotoxins produced as a secondary metabolite by toxigenic fungi in the genera aspergillus and penicillium. Ochratoxin A (OTA) is the most toxic and known to be mutagenic, carcinogenic, teratogenic and can also cause nephrotoxicity and hepatotoxicity [21]. It can be found in biological fluids, environmental samples, and foodstuffs. Various analytic tools are available for detection and quantification of OTA including chromatographic [22], RT-PCR [23], ELISA [24], and various versions of lateral flow immune assays [25]. Although the aforementioned methods are capable of offering good limits of detection, the high cost of instrumentation and difficulty of utilization hamper their use. Hence, there is a need for development of easy to use, cost effective, highly sensitive, and robust techniques for fast detection of OTA to avoid the negative health consequences that may occur.

In the past, Giraudi et al. and our group reported a new approach to detecting OTA using OTA binding peptides generated from a combinatorial library or phage display libraries and the peptides have similar binding profiles like aptamers [26,27,28,29,30]. Concerning the small hexapeptide SNLHPK developed by Giraudi et al., an affinity of ca. 3.4 x 10^4^ M^− 1^ for OTA was obtained after immobilization on standard solid substrates [30]. In this work, the goal is to develop a proof of concept, which could be the base of future biosensor platform development for toxin detection that uses cuvettes as a simple means of detection on site. In this study, the proof of concept is conducted by using ochratoxin A as a reference molecule and the hexapeptide SNLHPK as a biological recognition element. Therefore, it is necessary to immobilize the biological recognition element (a hexapeptide) on the surface of the PS based cuvette. Our objective is to create grafted selective and specific surfaces on a polystyrene substrate for the recognition of a specific toxin. Therefore, we have developed varied strategies to functionalize the polystyrene surface with various coupling agents including the siloxane derivative GPTS, divinyl sulfone (DVS), carbonyl diimidazole (CDI), and cyanogen bromide (CNB) for the first time. We have further investigated the structure and the functionality with the peptides after immobilization on activated polystyrene functionalized with these coupling agents. Figure 1 shows the scheme followed to functionalize the activated polystyrene.

## 2. Materials and Methods

(3-Glycidyloxypropyl) trimethoxysilane (GPTS,98%), triethyl amine (99%), cyanogen bromide (CNB, 97%), divinyl sulfone (DVS; 97%), carbonyl diimidazole (CDI, 90%), hydrochloric acid (37%), hydrogen peroxide (30%), sodium azide, citric acid, sulfuric acid (96%), bovine serum albumin (BSA), horseradish peroxidase (HRP), ochratoxin A (OTA) and sephadex ™ G25 were all purchased from Sigma Aldrich (France) and used without further purification. Phosphate buffered saline (PBS 10×) and 2,2′-azino-bis (3-ethylbenzothiazoline-6-sulphonic acid) (ABTS) chromogen/substrate solution for ELISA were purchased from Invitrogen. The peptides were synthesized by Smartox (France).

### 2.1. Surface Preparation

Polystyrene (PS) substrates/cuvettes were sonicated for 5 min in each solvent Isopropanol, methanol, and double deionized water. After drying, the PS substrates were soaked in a 6 M hydrochloric acid solution for 2 h. This was followed by a thorough wash with deionized water; the surfaces were soaked again in a freshly prepared Pirhana solution (3-part sulfuric acid and 1 part hydrogen peroxide) for 16 h. The substrates were then thoroughly rinsed with deionized water and dried with compressed air. Prior to functionalizing with the various biomolecule coupling agents the surfaces were kept within an ultraviolet/ozone probe and a decontamination (UV PSD-Novascan Technologies) chamber for 2 h. These subsequent treatments in acid and UV chambers ensured the OH termination of the surfaces, and they were immediately treated with the various coupling agents. GPTS functionalization: the OH terminated surfaces were soaked in a solution containing 5% (by volume) of GPTS in isopropanol. The surfaces were kept in a shaker overnight. The surfaces were thoroughly rinsed with isopropanol for 2 h to remove the excess adsorbed GPTS and then kept in an oven at 55 °C for 1 h prior to functionalizing with peptides. Cynogen (CNB) functionalization: 10% triethyl amine in isopropanol was added to the OH terminated surfaces, and the reaction was left shaking for 30 min. Then the surfaces were soaked in 0.75 M of cynogen bromide solution in dry isopropanol. The reaction was left stirring overnight. The modified surfaces were rinsed thoroughly with isopropanol and dried for immobilization of peptides. Divinyl sulfone (DVS) functionalization: Similar to the functionalization of CNB, the surfaces were treated with 10% (*v/v*) triethyl amine in isopropanol for 30 min followed by soaking in 5% (*v/v*) solution of DVS in isopropanol and the solution was left shaking overnight. The modified surfaces were thoroughly rinsed with isopropanol then dried for immobilization of peptides. Carbonyl diimidazol (CDI) functionalization: the OH terminated surfaces were treated with 10% (*v/v*) triethyl amine in isopropanol for 30 min, followed by soaking the surfaces in 0.5 M solution of CDI in isopropanol and the solution was left shaking overnight. The modified surfaces were thoroughly rinsed with isopropanol then dried for immobilization of peptides.

Peptide functionalization: to evaluate whether the various chemistries developed were amenable for use in biosensor formats, hexapeptide (SNLHPK), which was recently reported [25] to bind ochratoxin A, immobilized and perform competition ELISA. The terminal reactive groups in the various modified surfaces/cuvettes were treated with a solution containing 50 µg/mL hexa peptides and 50 mM PBS at pH 7.4 left shaking overnight at room temperature. The cuvettes were washed with wash buffer (PBS buffer containing 0.05% Tween; pH 7.4) to remove unbound peptides. The surfaces were subsequently treated with block buffer (PBS buffer pH 7.4 containing 0.1 BSA) for 1 h followed by washing with wash buffer.

### 2.2. Instrumentations

FT-IR, XPS, AFM, and contact angle measurements were performed to investigate the structures and confirm the covalent nature of immobilization. For all instrumentations analysis for each surface 3 triplicate measurements were performed. 

The Fourier-transform infrared spectroscopy (FTIR) spectra was recorded using a BrukerVertex 70 FT/IR Plus, a spectrometer with the help of a diamond-attenuated total reflectance (ATR) accessory was used. Typically, 128 scans at 2 cm^−1^ resolutions were recorded with cm^−1^ in the 4000–650−cm^−1^ region. During measurement, the cell compartment was flushed with dry nitrogen gas. For each surface, 3 triplicate measurements were performed.

The X-ray Photoelectron spectroscopy (XPS) measurements were recorded using a PHI ESCA-5500 spectrometer aluminum X-ray source. Samples were analyzed in an ultra-high vacuum (UHV) chamber pressure between 5 × 10^–9^ and 2 × 10^–8^ torr. 

Contact angle measurement: Static contact angles of DI water on the various glassy carbon substrates were measured using a KRUSS DSA30 instrument (KRUSS, Toulouse, France) equipped with an LED lighting unit, high-quality optical components, and a high-resolution camera. Measurements were made by delivering a 3 µL drop of Milli-Q water from a micro-syringe onto the surface of the sample mounted on an illuminated horizontal stage. The reported contact angle results are the average of ten measurements taken at different positions on each GC surface.

Atomic force microscopy (AFM): An Asylum MFP 3 D Infinity microscope (Oxford Instruments, Abingdon, United Kingdom) was used to perform AFM characterization of surfaces. Tests were carried out using a tapping mode, allowing to obtain the topography of sample surfaces. Images of 2 × 2 µm^2^ were then obtained using a silicon probe (AC160R3-TS from Asylum). Mean roughness (Ra) and root mean square roughness (Rq) of images were then measured for each sample surface. Images different zones of the surface were characterized for each sample in order to verify repeatability of results.

### 2.3. Synthesis of OTA HRP

The conjugation protocol was based on a modified version described by Xiao et al. 1995 [31], 5 mg of OTA dissolved in 500 µL of acetone was added to 7.5 mg of 1, 10-carbonyldiimidazole (CDI) and stirred in an amber vial for 20 min at ambient temperature. This reaction mixture was added drop by drop to an enzymatic solution of HRP (21 mg dissolved in 2 mL of carbonate buffer, pH 9.8), and this reaction was allowed to proceed for 4 h at ambient temperature under stirring and protected from light. The reacted OTA was separated from the unconjugated OTA by passing aliquots (500 mL) of this final reaction mixture through columns filled with sephadex™ G25, with a mobile phase of phosphate buffer saline (PBS, 15 mM, pH 7.4). Only the brownish fractions were collected for a total volume of 5.0 mL. Collected fractions were stored at 4 °C.

### 2.4. Peptide Based Enzyme Linked Immunosorbent Assay (ELISA) with Peptide SNLHPK in 96 Well Plate

Polystyrene white microtiter plate wells (Maxisorb LumiNunc, Thermoscientific, Boston, MA, USA), coated with the synthetic peptide SNLHPK at an optimized concentration of at 1 µg/mL in carbonate buffer (15 mM Na_2_CO_3_, 35 mM NaHCO_3_, 0.2 g/L NaN_3_, pH 9.6) were incubated at 37 1 C for 3 h. Non-specific binding sites of the peptide-coated wells were blocked with 5% nonfat dry milk in PBS containing 0.1% Triton X-100 (milk buffer) at room temperature (RT) for 3 h before performing the test. In each well, 1000 µg·L^−1^ of OTA-HRP (horseradish peroxidase) were added, combined with 50 mL of phosphate buffer saline supplemented with unlabeled OTA. The reaction was left for 30 min at RT.

After washing unbound OTA, ABTS Liquid Substrate 500 μL of substrate solution was added. It was incubated at room temperature for color development. For 25 min, color development was monitored with an ELISA plate reader at 405 nm. Equal volumes of stop buffer (0.01% sodium azide in 0.1 M citric acid) were added. The result obtained was inversely proportional to the concentration of unlabeled OTA. During each test, nonspecific binding (negative control) was determined by using an incubation mixture (OTA-HRP) in which the peptide SNLHPK was replaced by 100 mL of carbonate buffer. All the samples were tested in triplicate. In that direct competitive peptide-based ELISA, results are expressed in B/B0 dose logarithmic function. B and B0 represent the enzyme-bound activity measured in the presence or absence of competitor, respectively. The standard curve was traced by plotting standard concentrations on *x*-axis (logarithmic scale) and percentage of maximal binding (express in % of B/B0) on *y*-axis (B/B0 = f (log [OTA])). The binding values are obtained by dividing the absorbance at 405 nm of each testing well B (the absorbance measured when OTA-HRP and unlabeled OTA are in competition with SNLHPK peptide) by the absorbance of the positive control well B0 (maximum absorbance obtained with OTA-HRP). Decreasing exponential functions (y = y_0_ + A.e^−R0x^) were performed on the standard curve B/B0 = f (log [OTA]) by using Origin Lab software. The limit of detection (LOD) was obtained from the equation for y = A, the maximum with y = y_0_ and IC_50_ with y = 50.

### 2.5. Peptide Based Enzyme Linked Immunosorbent Assay (ELISA) in Cuvette

To check the functionality of the immobilized peptides, a direct ELISA has been performed. Ochratoxin A labelled with horseradish peroxidase (OTA-HRP) was synthesized and serially diluted to optimize the conditions to perform direct ELISA (OTA-HRP binds to the immobilized peptides). Capture peptide (SNLHPK) was diluted with PBS to a concentration of 1 µg/mL. One ml was immediately added to each of the activated cuvettes. The cuvettes were sealed with parafilm and incubated for overnight shaking at room temperature. The cuvettes were aspirated to remove liquid and washed 4 times with wash buffer (0.05% Tween 20 in PBS). Each wash consisted of adding 1 mL wash buffer, followed by aspiration. After the last wash, the cuvettes were dried on paper towels. 1 mL blocking buffer (1% BSA in PBS) was added to each cuvette and incubated for 1 h at R.T. The cuvettes were aspirated and washed 4 times with wash buffer. Serial dilutions of the prepared OTA-HRP were added to the cuvettes in diluent (0.05%Tween 20, 0.1% BSA in PBS) 500 μL was added per cuvette. It was incubated 30 min at room temperature and washed 4 times. To each cuvette, ABTS Liquid Substrate 500 μL of substrate solution was added. It was incubated at room temperature for color development. For 25 min, color development was monitored with an ELISA plate reader at 405 nm. Equal volumes of stop buffer (0.01% sodium azide in 0.1 M citric acid) were added. During each test, nonspecific binding (negative control) was determined by using a solution of unconjugated HRP or an incubation mixture (OTA-HRP) in which the peptide SNLHPK was replaced by 100 μL of PBS buffer. All the samples were tested in triplicate.

## 3. Results and Discussion

### 3.1. Surface Functionalization and Analysis

Polystyrene is resistant to covalent functionalization; however, treatment of the surface with either acid, base, or UV provide functionalities like COOH; OH, or NH_2_ depending on the various adjustments made to the procedure. In this study, as already described in the “methods” section, the treatment of PS with Piranha followed by UV plasma offers OH termination of the PS substrates. The OH functionalized substrates were treated with four different types of coupling agents for tethering short peptides. The various coupling agents studied in this work are well known for the functionalization of amine; carboxyl; thiol, and hydroxyl bearing biochemicals. Apart from the siloxane containing coupling agents (GEPTS), the others are rarely used to functionalize hydroxyl terminated polystyrene. In this work the coupling efficiency as well as utilization of the modified substrates for functionalization of peptides for detection of ochratoxine is demonstrated. 

The sulfone group in divinyl sulfone (DVS) is known to possess a strong electron withdrawing force to facilitate the formation of a bond with nucleophiles such as thiols, amines, and alcohols through Michael type addition reaction [32,33]. While in the case of CNB, the formation of imido carbonates because of activation of the OH terminated PS substrates facilitates the coupling of proteins in general through the reaction of primary amines of the protein. Reaction with the imido carbonate, on the contrary, due to the presence of water would deactivate the surface by forming carbamate groups on the PS-surfaces which are inert toward primary amines [34]. As in both DVS and CNB, the activation with CDI for OH terminated substrates proceeds in alkaline media (see method section) where the attack of the cabonyl group by the OH groups form imidazoyl cabamate, which itself has high reactivity to further react with primary amine of proteins to finally form a one carbon length and stable N-alkyl carbamate linkage [14]. The case of activation with siloxane derivative GEPTS was different; where no prior activation with alkaline media was required since the functionalization proceeds through formation of hydrogen bonds and baking in an oven at 50 °C for 1 h to form covalent bonds between O and Si [35]. The epoxy functional group was highly reactive to couple with tertiary amine groups on peptides. Due to the damage caused to PS surfaces by some organic solvents, products commonly used to dissolve GEPTS like toluene are not used to dissolve the GEPTS solution to activate the PS surface. GEPTS was also reported to form multilayers [36] in the presence of water. Hence, dry isopropanol and/or methanol were chosen as ideal solvents to functionalize the PS surfaces.

Peptide functionalization of the activated substrates varies with the coupling agents. For instance, DVS reacts with OH and NH_2_, the peptide used in this work bears the structure H-Ser-Asn-Leu-His-Pro-Lys-OH containing free OH, side chain ε-NH_2_ and secondary amine from histidine and NH_2_ end of the peptide. All are possible sites of functionalization varying orientation for maximizing the chance of toxin detection. However, the reaction lysine ε-NH_2_ is slow and incomplete, requiring very basic reaction conditions [37]. GPTS was capable of reacting directly with the amines of the peptide without prior activation with other activating agents like glutaraldehyde (which is used in the case of other siloxanes like APTS). The reactivity of GPTS is due to the reactive epoxide ring. While CNBr activated surfaces, the electrophilic carbon atom which forms imidocarbonate [38] reacts with the ε-NH_2_ of the lysine of the peptide under study as well as the N terminal of the peptide. Likewise, the lysine ε-NH_2_ is also involved in the coupling of the peptide with activated surface but the stability of the bond is compromised as it is not as strong as the bond with the N-terminal [39].

After modification of the surface, the contact angle of water drops on the various PS substrates provided rapid information on the nature of the modification hydrophilicity/hydrophobicity and extent of roughness. The water contact angle of the various modified PS substrates is described in Figure 2.

The contact angle of unmodified PS (94.9 ± 3) indicates the high level of hydrophobicity; it is the lowest in wettability compared to most modified substrates. Upon treatment with Piranha followed by UV-Plasma exposure; the substrates hydrophilicity and lowest wettability increased shown with a marked drop in contact angle (29.7 ± 3.1) compared to substrates obtained after further functionalization. This is attributed to the OH termination of the PS surface. Upon functionalization with the various coupling agents the contact angle varies depending on the C-H content of the group attached on the substrate. Unlike APTES modified substrates [11], where the substrates are eventually functionalized with amine bearing groups, the contact angle did not show an increase of more than the unmodified PS substrates. However, for GEPTS functionalized surfaces (101.2 ± 1.15) attained the higher hydrophobicity compared to the unmodified PS substrate due to the CH content which covers the surface (Figure 2). For other coupling agents an increase in contact angle compared to the OH functionalized substrates was also observed; PS-DV (67.87 ± 3.65), PS-CN (61.78 ± 2.41) and PS-CD (61.2 ± 0.6) but lesser in hydrophobicity compared to PS-GEPTS. This is attributed to the formation of hydrogen bonding in the case of PS-DV, PS-CN, and PS-CD. Upon functionalization of the substrates with hexapeptide, the extent of hydrogen bond formation is enhanced upon peptide functionalization of the surfaces where all the peptide modified substrates exhibit contact angle values all in the same range (Figure 2).A decrease in contact angle upon termination with OH or peptide groups can also be attributed to an increase in surface energy and surface roughness of the substrates. Moreover, the roughness of the modified substrate which was obtained from AFM measurements and the contact angle information revealed the immobilization of the short peptide [39].

The various elements and functional groups on the modified substrates have been studied by XPS. Core level O1s, N1s, C1s, Si1s, and S2p spectra of the various PS substrates are depicted in Figure 3.

Depending on the binding energies extracted from the XPS of the modified PS substrates, it is possible to verify the covalent functionalization of the substrates. When the PS substrates treated with piranha followed by UV-plasma changes of the nature of the carbons and the oxygens leading in the formation of carboxy, carbonyl, phenolic, and hydroxyl groups. The functional groups formed have characteristic binding energy extracted from the XPS spectra. In all the functionalized substrates the peak corresponding to C1s showed the main peak for the C-C sp2 peak (284.46 eV with fwhm 1.6 ± 0.3); C-C sp3 (285.1 eV); phenolic hydroxyl (286 eV); C atoms bonded to oxygen in hydroxyl configurations (286.5 eV); carboxylic C (288.6 eV) and carbonyl carbon (287.5 eV) [40,41]). Formation of the various functional groups also evidenced by the binding energy of the O1s peak centered at 532.16 eV which could be attributed to C=O (carbonyl and carboxylic 531.1 eV), C-O (hydroxyl and ethereal 532.1 eV) and C-O (phenolic 533.4 eV) [42]. 

Functionalization of piranha-UV treated substrates with solution of divinyl sulfone; the XPS spectra of PS-DV (Figure 3C) reveals binding energy at 168 eV which corresponds to the sulfone group [43,44]. Likewise, activation of the piranha-UV treated substrates with a solution of cynogen bromide (CNBr), (Figure 3B) the C1s from dioxo imine [-O-C(=NH)C-O-] were centered at 286.9 eV [45], and according to literature reports, the N1s peak comprises two main peaks with binding energies 398.5 and 399.9 eV corresponding to imine(N=C) and amine (NH) linkages [46]. After N, Nʹ-carbonyl diimidazol activation of the modified PS substrates the carbon from the carbamate [-O-(C=O)-N-] was centered at 289.5 eV, and the C-N was centered at 286.1 eV [47]. The N1s was explained similarly to the imine and amine NH similarly to the CNB. GPTS functionalization (Figure 3D) of the piranha–UV treated PS substrate revealed the characteristic Si2p peak centered at binding energy of 102 eV expected for triethoxysilane modified substrates [48]. 

XPS was used to detect the presence or absence of the protein on the surface. Wang et al. 2002 [49], used the N1s signal from the peptide bond of the protein to detect the adsorption of protein onto their plasma-treated membranes. As shown in Figure 3, there are increases in signal and broadness (fwhm) after coupling with the hexapeptide in the case of CDI and CNB based coupling, while there is the appearance of N1s peak in the case of GPTS and DVS coupling techniques, confirming the peptide functionalization of the substrates. Extracting the percentage of the C1s and N1s from the peptide immobilized surfaces similarity confirms the immobilization of the peptide SNLHPK [50]. 

Complementary results were obtained from the FT-IR of the various substrates (Figure 4).

Piranha –UV treated substrates show bands 1630 cm^−1^ and 3300–3500 cm^−1^ that can be attributed to stretching vibrations of OH groups [13,51]. The peaks 3300–3500 cm^−1^ also appear in peptide modified or coupling agent-modified substrates, which signify the whole PS surface was not fully utilized by the coupling agents or peptides, indicating residual OH groups. This region also corresponds to the hydrogen bonded N-H stretch on surfaces modified with peptide, CNBr, and CDI. In all substrates peak corresponding to symmetric/asymmetric stretching vibrations of CH_2_ and CH_3_ are observed around 2970–2870 cm^−1^. The C-O vibrational stretch is observed at 1120 cm^−1^. Stretching vibrations for nitrile appear in the region 2200–2100 cm^−1^ for CNB functionalized substrates. In addition to this attack of the nucleophilic carbon of CNB modified substrate results in the formation of imine group, which has a stretching vibration of approximately 1690–1650 cm^−1^. This region also corresponds to stretching vibrations of -C=C- of the imidazole in CDI functionalized substrates. The peak around 1680–1630 cm^−1^ corresponds to amide stretch of the peptide backbone and N-H deformation is observed around 1650–1510 cm^−1^. For DVS functionalized substrates peak at 1227 cm^−1^ and 1058 cm^−1^ correspond to the sulfone group [52]. Finally, peaks in the region 1465–1375 cm^−1^ were attributed to C-H and O-H deformations [53].

AFM was used to investigate the effect of piranha followed by UV plasma treatment and the nature of functionalization with various coupling agents order over the surface Figure 5.

Furthermore, AFM would also reveal the way peptides functionalized over the surface. AFM was performed in tapping mode in the air, and the images were realized in an area of 2 × 2 µm^2^. In addition to the figures, the root mean square surface roughness (Ra) was calculated for the various substrates. There was a decrease in Ra up on Piranha-UV treatment of the PS substrates from 2.93 nm to 2.42 nm, revealing the mild surface treatment of the surface using piranha-UV plasma combination as has been reported for nitrogen plasma by M.J-Wang et al. 2005 [54]. Upon functionalization with the various coupling agents the roughness varied differently: there was no change compared to the piranha–UV plasma modified surfaces in the case of modified surfaces, PS-GP (2.41 nm); a decrease in Ra for DVS modified substrates, PS-DV (1.48 nm); a slight increase in Ra for CDI modified surfaces, PS-CD (3 nm) and cynogen modified surfaces, PS-CN (3.47 nm). After treatment with solution of the hexapeptide, in all cases the Ra increased PS-GP-pept (4.04 m); PS-DV-pept (8.23 nm); PS-CN-pept (7.65 nm); PS-CD-pept (3.42 nm).

### 3.2. Assay Optimization for Detection of Ochratoxin A

Peptide based enzyme linked immunosorbent assay (ELISA) in 96-well plate: The synthetic SNLHPK peptide was tested in competitive ELISA in 96-well plates to verify its functionality for OTA detection. This competitive ELISA was performed with a concentration in peptide SNLHPK at 1 µg/mL for a concentration of OTA-HRP labeled at 1000 µg·L^−1^. The concentration of OTA varied from well to well (0 µg·L^−1^, 0.25 µg·L^−1^, 0.5 µg·L^−1^, 1 µg·L^−1^, 2 µg·L^−1^, 5 µg·L^−1^, 10 µg·L^−1^ and 25 µg·L^−1^). The competition proceed is represented by the exponential curve fit for the standard OTA in PBS. The calibration curve in PBS (Figure 6) shows that inhibition starts at 0.5 µg·L^−1^.

The maximum inhibition is obtained at 10 µg·L^−1^, and half inhibition occurs at 2.1 µg·L^−1^. Inhibition is complete, which is expected since the tracer is also OTA-based. From the calibration curve parameters, and with the SD values, the range of detection of OTA in PBS is between 0.5 and 10 µg·L^−1^.These results confirm that the synthetic peptide SNLHPK is active in the detection of OTA.

Assay optimization for binding of Ochratoxin A in cuvette: As described earlier, the orientation of the peptides on the surface varies depending on the reaction of the peptide with the surface anchored coupling agent. In the case of surfaces functionalized with CDI and GPTS, the hexapeptide (SNLHPK) reacts with the N-terminal of the peptide as well as the primary NH_2_ of the lysine resulting in the formation of an amide bond [-(C=O) NH-] for CDI and epoxy ring opening forming [-N CH_2_C(OH)CH_2_-] for GPTS. Likewise, CNB functionalized substrates react with the N-terminal NH_2_ group of the peptide and the free NH_2_ group of the lysine through the formation of imidocarbonates. In addition, the imidazole group of the histidine may also involve a reaction with the imidocarbonate. In the case of DVS functionalized substrates, the coupling reaction proceeds in a similar way through the imidazole of the His as well as the primary amines of the peptide (N-terminal and the lysine NH_2_). Unlike the peptide, NFO_4_ [26,28] orientations of the hexapeptide under study are not reported to affect the OTA binding. Due to the covalent nature of surface functionalization, strategies followed in this study provide sensor formats with analytical performance, high stability, and highly reproducible results. As shown in Figure 7, the plateau of the sigmoidal curve reached with a concentration of OTA-HRP in the range of 28.5–32.7 µg/mL. 

## 4. Conclusions

Various surface chemistries for the immobilization of peptides were demonstrated based on ease of functionalization, stability, and selectivity for the detection of ochratoxin A. The modified surfaces were characterized by XPS, AFM, ATR FT-IR and contact angle measurements. In all the chemistries, controlled functionalization of the peptides was achieved. Labelled horseradish peroxidase (OTA-HRP) was synthesized to perform ELISA on the various modified cuvettes to confirm the functional chemistries utilized to anchor the peptides. The addition, the acid 2,2′-azino-bis(3-éthylbenzothiazoline-6-sulphonique) (ABTS) produced a blue color. The immunosensor response was fitted to a sigmoidal curve with r^2^ > 0.99 (*n* = 3) in all cases. The ELISA binding assay response obtained was found to be in the same order for all the chemistries studied. The results also show an efficient and reliable method for functionalizing polymer-based surfaces that could lead to the development of very cheap and reliable biosensors. However, it is necessary to determine the optimal conditions for the development of a peptide based competitive enzyme linked immunosorbent assay (ELISA) in cuvette (determination of the standard curves). The surface functionalization approaches can be extended to small antibodies, aptamers, and nanoenzymes such as advanced ELISA systems reported by Z Lyu et al. [55] to produce future stable and extremely low-cost biosensors. 

## Figures and Tables

**Figure 1 sensors-22-09538-f001:**
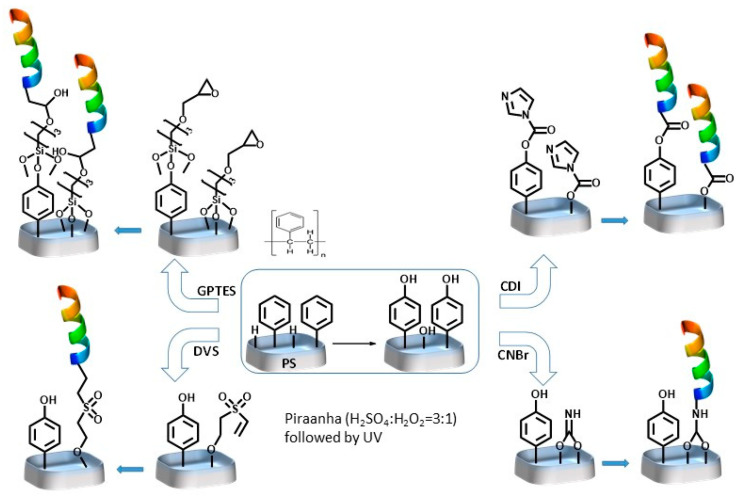
Schematic representation of peptide functionalization of polystyrene cuvettes.

**Figure 2 sensors-22-09538-f002:**
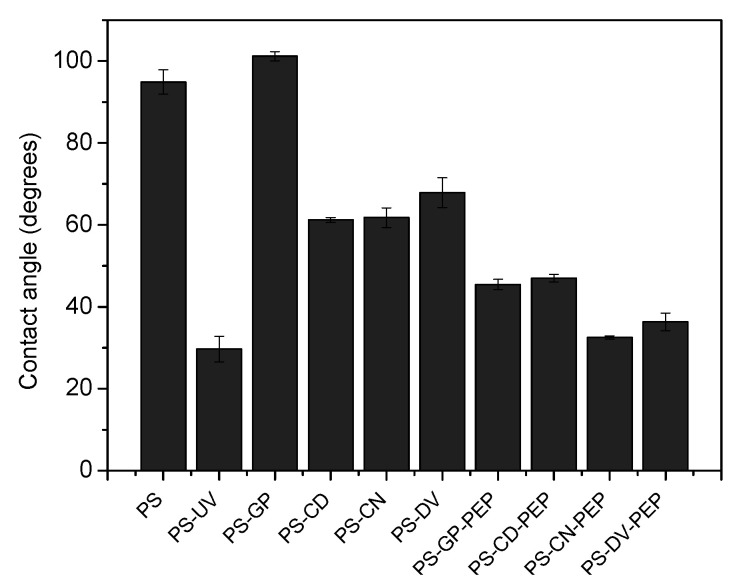
Water contact angle of PS substrates before and after functionalization.

**Figure 3 sensors-22-09538-f003:**
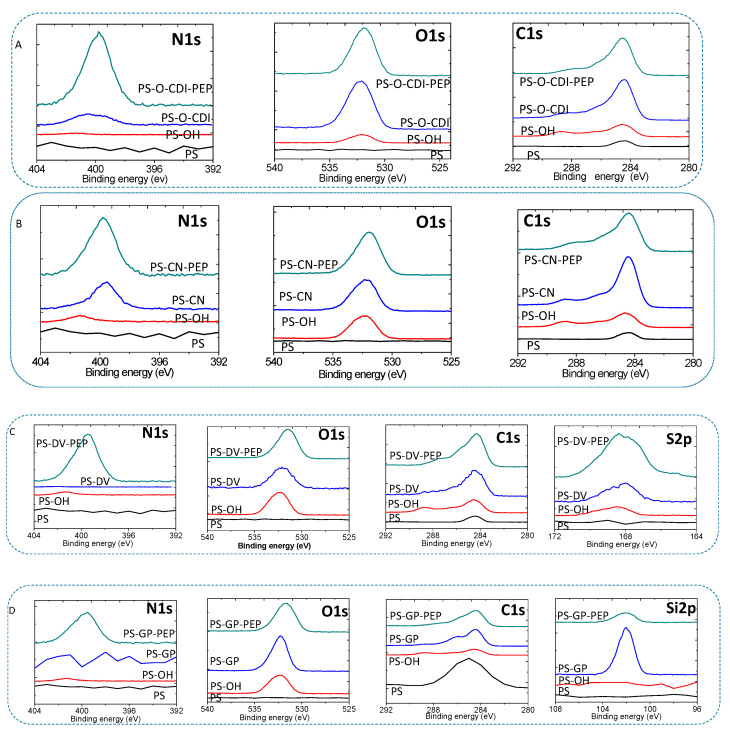
The XPS spectra depicts the N1s, O1s, C1s, and S2p core levels for the various substrates functionalized by the methods under study. (**A**) series represent the XPS spectra of –CDI-PEP modified PS. (**B**) are the XPS spectra of –CN-PEP modified PS. (**C**) are the XPS spectra of –DV-PEP modified PS. (**D**) are the XPS spectra of PS GP modified PS.

**Figure 4 sensors-22-09538-f004:**
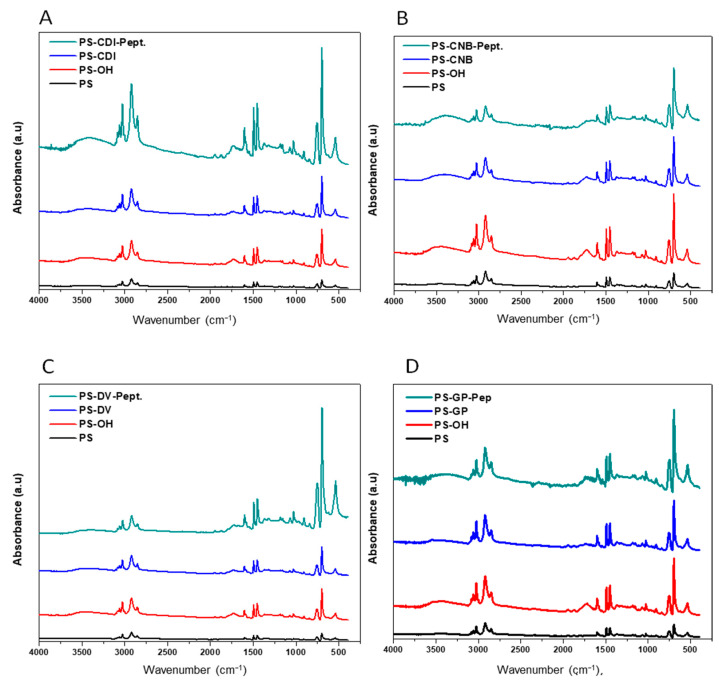
The ATR FT-IR of the various PS surfaces following activation with UV piranha, coupling agents and peptide coupling: (**A**) CDI modified PS. (**B**) CNB modified PS. (**C**) DVS modified PS. (**D**) GPTS modified PS.

**Figure 5 sensors-22-09538-f005:**
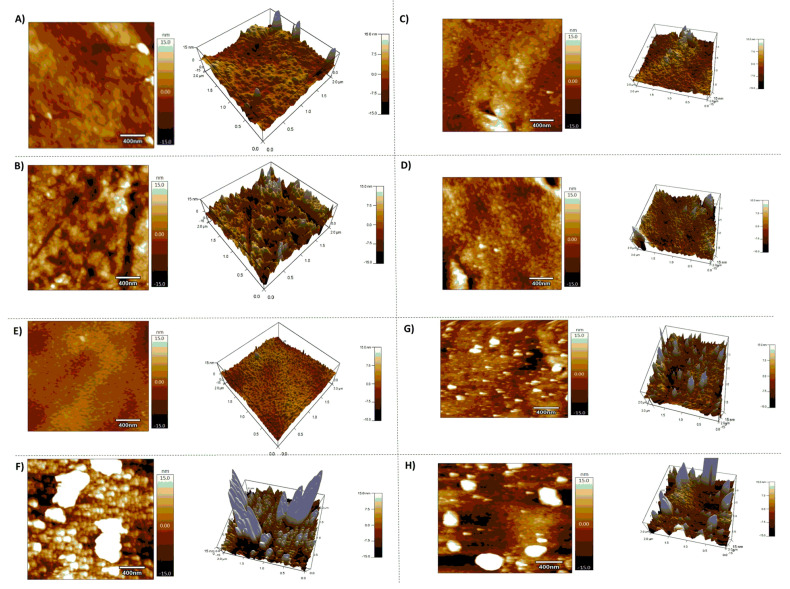
Atomic force micrographs (scale 2 × 2 µm^2^) after immobilization of the various PS-coupling agents and peptide coupling: (**A**) GPTS modified PS, (**B**) PS-GP-Pept. (**C**) PS-CDI (**D**) PS-CDI-Pept. (**E**) PS-DVS (**F**) PS-DV-Pept. (**G**) PS-CNB and (**H**) PS-CN-Pept. The images on the right in each figure corresponds to 3 D views.

**Figure 6 sensors-22-09538-f006:**
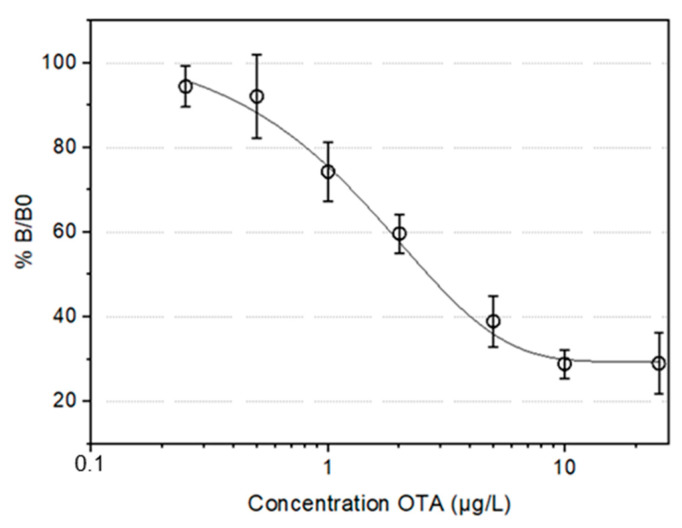
OTA calibration curves in PBS on 96-well plate. Competitive ELISA for the detection of OTA in PBS were performed with a concentration of peptide SNLHPK (1 µg/mL) for a concentration of OTA-HRP labeled at 1000 µg·L^−1^. B and B0 represent the bound enzyme activity measured in the presence or absence of competitor respectively. Each point is the average ± standard deviation of three independent assays each with 4 measurement (*n* = 12).

**Figure 7 sensors-22-09538-f007:**
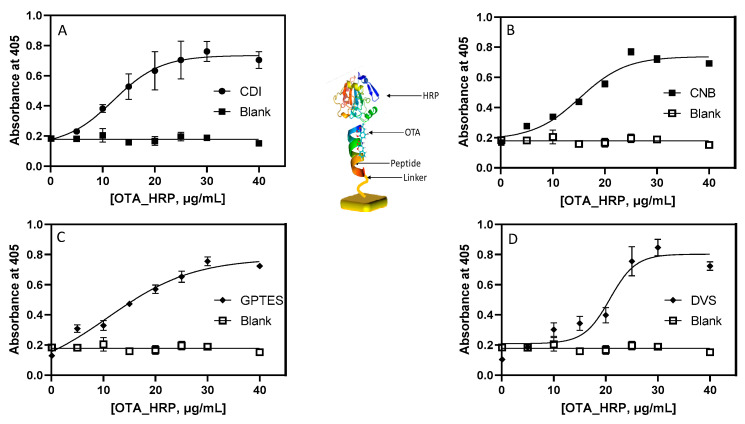
Binding curve for serial dilutions OTA-HRP (in 10 mM PBS pH 7.4) to cuvettes functionalized with 2 mg/mL of the peptide SNLHPK. (**A**) CDI modified PS. (**B**) CNBr modified PS. (**C**) GPTS modified PS. (**D**) DVS modified PS. In all experiments, the blank is the solution of unconjugated HRP.

## Data Availability

Not applicable.

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
