# Peer review of "Surface Functionalization Strategies of Polystyrene for the Development Peptide-Based Toxin Recognition"

_sensors, 2022, doi:10.3390/s22239538_

Round 1
Reviewer 1 Report
The article looks more like a statement of intent than an evidence-based study. The authors' statements are too general and ambitious, while only the possibility of creating modified layers on a polystyrene substrate is shown. Each specific application of such modified flexible films must be thoroughly proven. If we accept the material as a pilot and demonstrate the possibility of such an approach – to create grafted selective and specific surfaces on a polystyrene substrate, then it can be accepted. The goal is to develop a biosensor platform that uses cuvettes as a simple means of on-site detection.
However:
The chemistry of the process is well described, it is proved by various methods that the chemical structure changes on the surface of the PS. But the rest of the results are presented in insufficient volume and cannot confirm that the potentially developed material can perform the function of a sensor in place with a visual response. The data is clearly not enough.
But the approach to immobilize the surface turned out to be successful.
The process of creating new selective polystyrene surfaces is long and multi-stage. There is no data on reproducibility of surface properties in new (serial) productions. The methods of determining the edge wetting angle and near infrared spectroscopy are the most informative here.
There is little information and discussion of the results even on model solutions. Therefore, claims about the determination of mycotoxins in food and water are premature.
Author Response
"Please see the attachment."

Reviewer 2 Report
Obs.1.
The introduction chapter is very telegraphic and lacks fluency. I strongly recommend it be rewritten. Last paragraph I think belongs to ”Materials and Methods” section.
Obs.2.
There are a large number of odd expressions and grammatical mistakes. It will be of great importance if the proposed paper is reviewed by a native English speaker (for example, the text from lines 134-136 makes no sense, text from lines 199-201).
Obs.3.
Please introduce space between the values and the measurement units through entire manuscript.
Please use subscript for different substances, such as NH2 (and not NH2) and for measurement units.
Figures must be introduced after their first mention in the text.
Please check if the way of writing the references, without entering the name of the quoted article, is in accordance with the requirements of the journal.
Obs.4.
Please increased the size and resolution of Fig.4.
” Wavenumber” is spelled as a single word.
A large number of peaks are not presented or taken into discussion. For example, in Fig.4., peaks from 3000 cm-1 (CNB) , why did you not attribute them?
Author Response
"Please see the attachment."

Reviewer 3 Report
This work used the peptide to prepare a novel ELISA for toxin detection. The whole work looks good but still needs to provide more important information before publishing.
1. A real sample detection should be provided. For example, spike recovery experiment.
2. A commercial ELIAS should be used to compare the sensitivity of peptide one. Also, more related published advanced ELISA works towards toxins should be included for comparison.
3. For the XPS characterization, the N 1s spectra should be deconvoluted to prove the successfully immobilized peptide.
4. More advanced ELISA works should be included, like doi.org/10.34133/2020/4724505.
Author Response
"Please see the attachment."

Round 2
Reviewer 1 Report
The revised version of the manuscript is more consistent with its strategy, materials and title. The materials are supplemented and demonstrate the properties of the new systems.
Reviewer 2 Report
I have no other comments.
Reviewer 3 Report
The authors made good revisions, no comments anymore.